# Histone gene replacement reveals a post-transcriptional role for H3K36 in maintaining metazoan transcriptome fidelity

Michael P Meers[1,2], Telmo Henriques[3†], Christopher A Lavender[4], Daniel J McKay[1,2,5,6], Brian D Strahl[7,8], Robert J Duronio[1,2,5,6,8], Karen Adelman[3†], A Gregory Matera[1,2,5,6,8]*

[1]Curriculum in Genetics and Molecular Biology, The University of North Carolina at Chapel Hill, Chapel Hill, United States; [2]Integrative Program for Biological and Genome Sciences, The University of North Carolina at Chapel Hill, Chapel Hill, United States; [3]Epigenetics and Stem Cell Biology Laboratory, National Institute of Environmental Health Science, Durham, United States; [4]Integrative Bioinformatics Support Group, National Institute of Environmental Health Science, Durham, United States; [5]Department of Genetics, The University of North Carolina at Chapel Hill, Chapel Hill, United States; [6]Department of Biology, The University of North Carolina at Chapel Hill, Chapel Hill, United States; [7]Department of Biochemistry and Biophysics, The University of North Carolina at Chapel Hill, Chapel Hill, United States; [8]Lineberger Comprehensive Cancer Center, The University of North Carolina at Chapel Hill, Chapel Hill, United States

*For correspondence: matera@unc.edu

Present address: †Department of Biological Chemistry and Molecular Pharmacology, Harvard Medical School, Boston, United States

**Abstract** Histone H3 lysine 36 methylation (H3K36me) is thought to participate in a host of co-transcriptional regulatory events. To study the function of this residue independent from the enzymes that modify it, we used a 'histone replacement' system in *Drosophila* to generate a non-modifiable H3K36 lysine-to-arginine (H3K36R) mutant. We observed global dysregulation of mRNA levels in H3K36R animals that correlates with the incidence of H3K36me3. Similar to previous studies, we found that mutation of H3K36 also resulted in H4 hyperacetylation. However, neither cryptic transcription initiation, nor alternative pre-mRNA splicing, contributed to the observed changes in expression, in contrast with previously reported roles for H3K36me. Interestingly, knockdown of the RNA surveillance nuclease, Xrn1, and members of the CCR4-Not deadenylase complex, restored mRNA levels for a class of downregulated, H3K36me3-rich genes. We propose a post-transcriptional role for modification of replication-dependent H3K36 in the control of metazoan gene expression.

## Introduction

Eukaryotic genomes function within the context of chromatin fibers composed of nucleosome units, each of which contains roughly 147 bp of DNA wrapped around a single histone octamer composed of two pairs of heterodimers (histone H2A-H2B, and H3-H4) (*Luger et al., 1997*). Histones are decorated with an array of covalent post-translational modifications (PTMs) that have been proposed to demarcate distinct chromatin domains in the genome (*Kharchenko et al., 2011*; *Rice et al., 2003*; *Schneider et al., 2004*; *Sullivan and Karpen, 2004*). The 'histone code' hypothesis posits that PTMs

**eLife digest** In a single human cell there is enough DNA to stretch over a meter if laid end to end. To fit this DNA inside the cell – which is less than 20 micrometers in diameter – the DNA is tightly wrapped around millions of proteins known as histones, which look like "beads" along a "string" of DNA. These histones can prevent other proteins from binding to DNA and activating specific genes. Therefore, cells use enzymes to chemically modify histones to allow particular stretches of DNA to be unwrapped at specific times.

Proteins are made up of building blocks called amino acids. A specific amino acid on histones known as H3K36 is modified in certain sections of DNA that suggest it affects the activities of many genes. However, the precise role of this amino acid remains unclear. Previous studies have tried to investigate this by removing the enzymes that modify it, but these enzymes can also modify many other proteins, making it difficult to know what exactly causes the changes in gene activity.

Fruit flies are often used in experiments as models of how genetic processes work in humans and other animals. Like us, fruit flies also package their DNA using histones. To investigate the role of H3K36, Meers et al. generated a mutant fruit fly that has a version of the amino acid that cannot be chemically modified by the normal enzymes. Unexpectedly, the experiments suggest that some changes in gene activity that have been previously reported to be caused by modifying H3K36 might actually be due to other factors. Meers et al. found that H3K36 modifications may instead "mark" certain genes to be more active than they otherwise would be.

These findings provide a starting point for understanding exactly how H3K36 regulates gene activity. The next challenge is to refine our understanding of how H3K36 modification affects genes in cancer and other diseases, which may aid the development of new therapies to treat these conditions.

play crucial roles in controlling gene expression by adapting the local chromatin packaging environment and recruiting structural or catalytic binding partners to confer or deny access to transcriptional machinery (*Bannister and Kouzarides, 2011*; *Jenuwein and Allis, 2001*; *Rothbart and Strahl, 2014*; *Strahl and Allis, 2000*; *Taverna et al., 2007*). Partly on the basis of this model, PTMs have been considered strong candidates for primary carriers of epigenetic information that contribute to cell fate specification during development (*Margueron and Reinberg, 2010*). This concept has been extended to suggest PTM dysregulation as a likely contributor to diseases characterized by altered gene expression and cell identity (*Chi et al., 2010*; *Lewis et al., 2013*).

In multicellular eukaryotes, support for the histone code hypothesis is largely based on phenotypes observed from studies in which the 'writer' enzymes responsible for catalyzing histone PTMs were inhibited or ablated. However, such experiments cannot rule out the possibility that these enzymes have other non-histone substrates, or play other non-catalytic (e.g., structural) roles, that confound analysis and assignment of observed phenotypes to the PTMs themselves. Several recent studies have employed a direct replacement of the endogenous, replication-dependent histone gene cluster in *Drosophila melanogaster* with transgenic clusters encoding non-modifiable mutant histones (*Graves et al., 2016*; *Günesdogan et al., 2010*; *Hödl and Basler, 2012*; *McKay et al., 2015*; *Pengelly et al., 2013*; *Penke et al., 2016*). This approach has enabled the deconvolution of phenotypes specific to histone PTMs from those specific to their writers. These studies have elucidated the relationship between PTMs and their writers, both confirming (*Pengelly et al., 2013*) and refuting (*McKay et al., 2015*) previously reported roles for certain residues on the basis of their corresponding writer mutant phenotypes. The approach also affords an opportunity to directly interrogate the function of other well-characterized histone PTMs for which a variety of functional roles have been described.

In contrast with many PTMs whose spatial distribution is skewed towards promoters and the 5' regions of genes, H3K36 di- and tri-methylation (H3K36me2/3) are enriched in coding regions and toward the 3' end of actively transcribed genes (*Bannister et al., 2005*). These marks are also preferentially enriched over exons as opposed to introns (*Kolasinska-Zwierz et al., 2009*). This distribution pattern suggests that H3K36me interfaces with RNA polymerase and contributes to transcription

elongation and/or RNA processing, rather than affecting gene expression via chromatin packaging at promoters. Indeed, H3K36me2/3 is known to suppress cryptic transcription initiation from coding regions in *Saccharomyces cerevisiae* by recruiting a repressive Rpd3 deacetylase complex to sites of active elongation (*Carrozza et al., 2005*; *Keogh et al., 2005*). It is also implicated in suppressing active incorporation of acetylated histones via histone exchange (*Venkatesh et al., 2012*). In cultured cells, ablation of human SETD2, which catalyzes H3K36 trimethylation, is suggested to alter a number of exon inclusion events by recruiting RNA binding proteins (*Luco et al., 2010*; *Pradeepa et al., 2012*). Conversely, H3K36me3 distribution across gene bodies is itself sensitive to perturbations in splicing (*de Almeida et al., 2011*; *Kim et al., 2011*). In addition to its role in transcription and RNA processing, a range of other activities have been attributed to H3K36me, including X-chromosome dosage compensation (*Larschan et al., 2007*), DNA damage response (*Jha and Strahl, 2014*; *Li et al., 2013*; *Pai et al., 2014*; *Pfister et al., 2014*), and three dimensional chromosome organization (*Evans et al., 2016*; *Smith et al., 2013*; *Ulianov et al., 2016*). However, to date, none of these putative roles for H3K36me have been evaluated directly in an H3K36 mutant animal.

Here, we report a comprehensive analysis of H3K36 function, focused on differential gene expression, transcription initiation, and chromatin accessibility phenotypes in transgenic *Drosophila* whose entire complement of replication-dependent H3 genes has been mutated to arginine at lysine 36 (H3K36R). Arginine approximates the charge and steric conformation of lysine, but cannot be targeted by lysine methyltransferases, and therefore represents an appropriate mutation with which to study the PTM-specific functions of H3K36. Although arginine is a conservative amino acid change, it also enables hydrogen bonding modalities that are distinct from those of lysine. In principle, in addition to phenotypes resulting from loss of H3K36 methylation, such a change could also result in other hypomorphic (partial loss of function) or neomorphic (gain of function) phenotypes.

In H3K36R mutants, we observed a decrease in the steady-state levels of highly expressed RNAs concomitant with increased transcription and productive expression from a variety of low-usage promoters. Though mutants exhibited bulk increases in histone acetylation, chromatin accessibility did not appreciably change at promoters. Surprisingly, we found that previously reported roles for H3K36 methylation, including suppression of transcription initiation in coding regions and regulation of alternative splicing, are not supported in *Drosophila* by transcription start-site (TSS) and poly-A RNA-seq analyses, respectively. Intriguingly, we found that certain genes are downregulated in H3K36R mutants but are rescued to wild-type levels by depletion of the Xrn1 exonuclease pacman, or the CCR4-Not deadenylase subunits, twin and Pop2. We posit a model whereby H3K36 methylation contributes to transcript fitness in order to maintain global transcriptome fidelity.

## Results

### H3K36R mutation causes widespread dysregulation of the transcriptome

We utilized a bacterial artificial chromosome (BAC)-based histone gene replacement platform (*McKay et al., 2015*) to generate *Drosophila* bearing a K36R substitution mutation in each of its replication-dependent histone H3 genes. Using this system, the endogenous histone gene cluster was deleted and complemented by a transgenic array of 12 copies of the native 5 kb histone gene repeat (*Figure 1*). As previously reported, H3K36R (K36R) mutants pupate at significantly reduced frequency compared to histone wild type (HWT) control animals, and fail to eclose into adults with 100% penetrance, despite exhibiting no obvious cell proliferation defects (*McKay et al., 2015*). Given the postulated role for H3K36 modification in co-transcriptional gene regulation, we sought to comprehensively compare the transcriptomic landscapes of HWT and K36R animals. We sequenced poly-A selected RNA, rRNA-depleted nuclear RNA, nucleosome depleted DNA (via ATAC-seq [*Buenrostro et al., 2013*]), and short, nascent, capped RNAs (*Henriques et al., 2013*; *Nechaev et al., 2010*) from third instar larvae. Collectively these methods interrogate the major steps in mRNA biogenesis (*Figure 1*).

We hypothesized that the K36R mutation would conform to a 'cis-acting,' direct model, wherein effects are confined primarily to genes containing high levels of H3K36me3. However, when we analyzed genome-wide differential expression from poly-A RNA and stratified genes by the chromatin 'states' in which they reside (as defined in *Kharchenko et al., 2011*), gene expression changes were

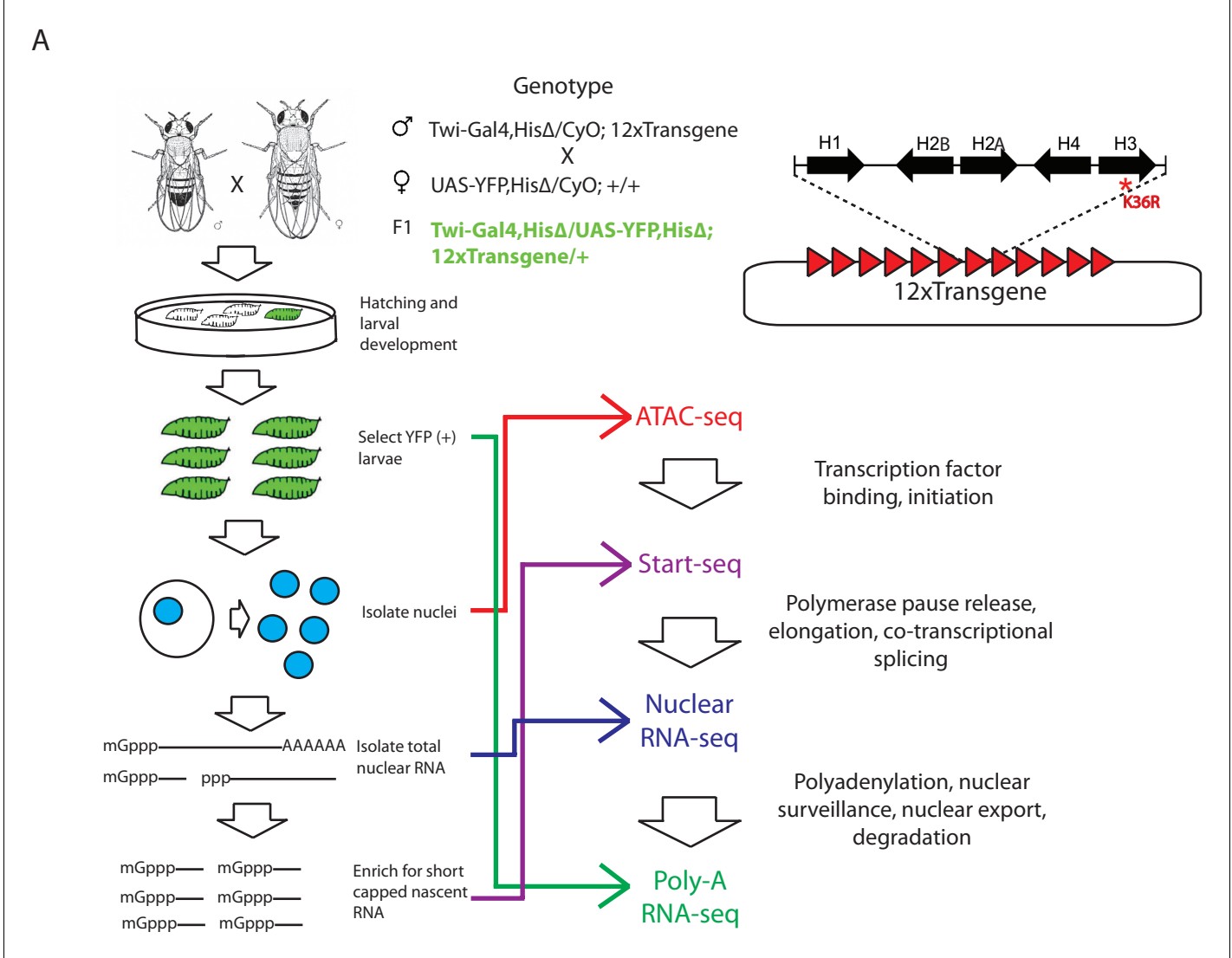

**Figure 1.** Strategy for interrogating the transcriptomic life cycle of H3K36R animals. (**A**) Schematic of experimental high-throughput sequencing methods applied to H3K36R animals. Twelve tandem copies of the histone repeat unit were cloned into a custom BAC vector and site-specifically integrated into the *D. melanogaster* genome as described in *McKay et al. (2015)*. Poly-A-selected RNA was sequenced from whole third instar larvae, ATAC-seq and rRNA-depleted nuclear RNA-seq were carried out from nuclei isolated from third instar larvae, and short, nascent, capped RNAs were selected from nuclei and subjected to 'Start-seq' (*Henriques et al., 2013*).

not confined to states characterized by high levels of H3K36 methylation (*Figure 2—figure supplement 1A*, states 1–4). Instead, when we stratified genes by H3K36me3 density (www.modencode. org), the mark was anticorrelated with gene expression change across the entire spectrum of H3K36me3 density, and largely uncorrelated with other methyl-states of H3K36 (*Figure 2A*, *Figure 2B*). Genes with high H3K36me3 density tended to decrease expression in K36R animals, whereas genes with low H3K36me3 density tended to increase expression in K36R animals. This finding suggests a global role for H3K36me in regulating gene expression, but one that is not confined to H3K36me3-rich loci, and therefore argues against an exclusively direct, local effect.

Because H3K36me3 is catalyzed co-transcriptionally (*Kizer et al., 2005*), and should therefore track roughly with gene expression, we also took the alternate approach of determining whether gene expression changes in K36R were correlated with the amount of expression normally observed in HWT. When we plotted differential expression against a specific transcript's HWT level, we found

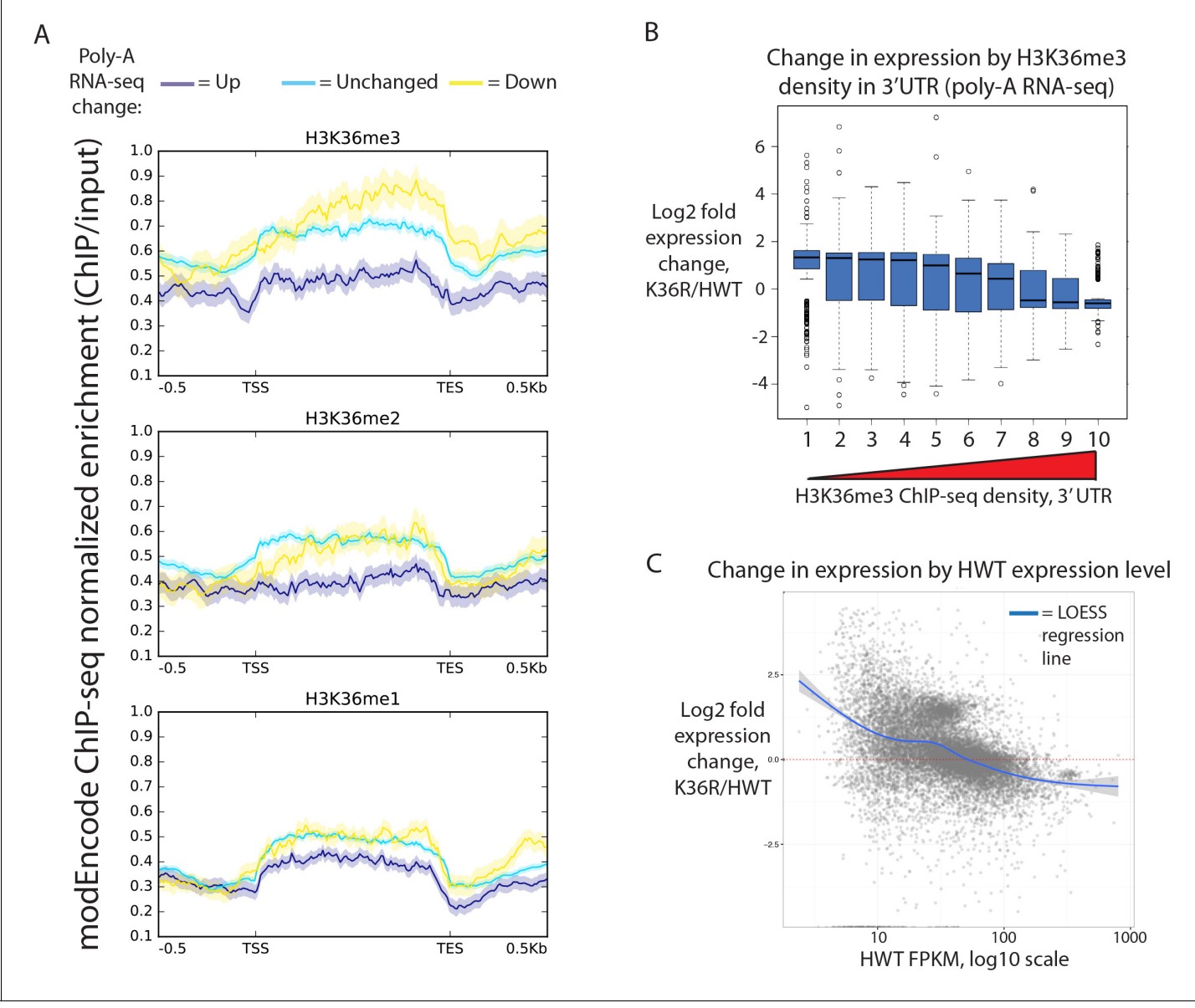

**Figure 2.** Transcriptome dysregulation in H3K36R mutants is correlated with H3K36me3 ChIP-seq. (**A**) Metagene plot describing the density of H3K36me3 (top), H3K36me2 (middle), and H3K36me1 (bottom) ChIP-seq across genes that are upregulated (purple), unchanged (blue), or downregulated (yellow) in H3K36R mutants as compared with HWT controls. (**B**) Boxplot of differential expression of gene cohorts stratified by density of H3K36me3 signal in the 3' UTR (1=lowest density decile, 10=highest decile). (**C**) MA plot with accompanying LOESS regression line plotting log2 fold change (y-axis) vs. HWT FPKM (x-axis) interpreted from poly-A RNA-seq data.

The following figure supplement is available for figure 2:

**Figure supplement 1.** Gene expression changes in H3K36R mutants.

that the effects of the K36R mutation were consistently anticorrelated with a gene's HWT expression level. That is, genes that were normally silent or lowly-expressed in HWT larvae experienced the largest relative increases in expression in K36R mutants, and highly expressed genes were preferentially reduced in K36R (*Figure 2C*, *Figure 2—figure supplement 1B*). RT-qPCR validation of select transcripts confirmed this observation, arguing against the likelihood of bias due to normalized RNA

input (*Figure 2—figure supplement 1C*). These results indicate that H3K36me-dependent expression changes could be caused by both direct (locus-specific) and indirect (locus-independent) effects.

## H3K36 mutants exhibit increased histone acetylation, but unchanged global chromatin accessibility

H3K36 methylation status has the potential to affect other histone PTMs, most notably H4 acetylation (H4ac) (*Carrozza et al., 2005*; *Keogh et al., 2005*) and H3K27 trimethylation (H3K27me3) (*Lu et al., 2016*; *Yuan et al., 2011*). This form of histone 'crosstalk' might contribute to the observed gene expression phenotypes. To formally evaluate this possibility, we assayed bulk levels of H4ac and H3K27me3 by western blotting. H3K27me3 levels were slightly reduced in H3K36 mutants (*Figure 3A*, *Figure 3—figure supplement 1A*), however characteristic polycomb target genes were largely unaffected (*Figure 2—figure supplement 1A*, *Figure 3—figure supplement 1B*). In contrast, H4ac levels were robustly increased (*Figure 3A*, *Figure 3—figure supplement 1A*), confirming the previously identified link between H3K36me and H4ac (*Carrozza et al., 2005*; *Keogh et al., 2005*).

To assay the spatial distribution of H4ac, we stained polytene chromosomes with an H4K12ac antibody. In both HWT and K36R mutants, we found that H4K12ac intensity was anticorrelated with DAPI bright bands (*Figure 3—figure supplement 1C*). The DAPI bright regions are thought to correspond to more transcriptionally silent chromatin. Therefore, the observed hyperacetylation in K36R mutants occurs in the more actively transcribed (DAPI dark) regions, consistent with previous observations (*Carrozza et al., 2005*; *Keogh et al., 2005*). Given these findings, we initially hypothesized that H4 hyperacetylation might contribute positively to chromatin accessibility in promoter proximal regions of genes that are upregulated in H3K36 mutants. To investigate this possibility, we carried out open chromatin profiling (ATAC-seq) and correlated these data with our differential expression (RNA-seq) analysis. Wild-type H4 acetylation density was also calculated using H4K16ac ChIP-seq datasets obtained from the modEncode consortium. As shown in *Figure 3—figure supplement 1D*, genes with the lowest levels of H4K16ac at their predicted promoters increased their expression levels in K36R mutants.

To localize open chromatin changes specifically to bona-fide sites of transcription initiation, we performed 'Start-seq', which precisely determines transcription initiation events by capturing nascent RNAs associated with early elongation complexes (*Henriques et al., 2013*; *Nechaev et al., 2010*). We adapted the protocol to isolate short, nascent, capped RNA from nuclei purified from third instar larvae (see Materials and methods). As shown in *Figure 3—figure supplement 2A–C*, Start-seq signal maps faithfully and robustly, with base-pair resolution, to annotated (observed) transcription start sites (obsTSSs), and demarcates sites of high nuclear RNA-seq. ATAC-seq signal accumulates most robustly in a window spanning roughly 150 nt upstream, and 50 nt downstream, of obsTSSs (*Figure 3—figure supplement 2D*). When we quantified HWT and K36R ATAC-seq signal from such a window surrounding all obsTSSs, we found that global changes in open chromatin were minimal between HWT and H3K36R animals (*Figure 3B*). Furthermore, changes in ATAC-seq at obsTSSs and differential expression in their corresponding genes was largely uncorrelated, with a large proportion of genes exhibiting differential expression changes independent of increased chromatin accessibility (*Figure 3C*). These results indicate that chromatin remodeling at promoters is not a major contributor to the observed global gene expression changes.

## Cryptic transcription initiation does not contribute to gene expression changes in H3K36 mutants

Given that increases in H4 acetylation in response to loss of H3K36me were thought to promote cryptic transcription in *S. cerevisiae* (*Carrozza et al., 2005*; *Keogh et al., 2005*), we evaluated potential cryptic initiation phenotypes in *Drosophila* H3K36 mutants. The consistent accumulation of Start-seq signal at bona-fide transcription initiation sites (*Figure 3—figure supplement 2A*) shows that this method is particularly ideal for identifying novel initiation elsewhere in the genome. By quantifying Start-seq signal at loci outside of annotated start-sites (obsTSSs), we identified thousands of _n_ovel _u_nannotated TSSs (nuTSSs) spread throughout the HWT genome, including a large proportion located within H3K36me3-enriched exons (*Figure 4A–B*).

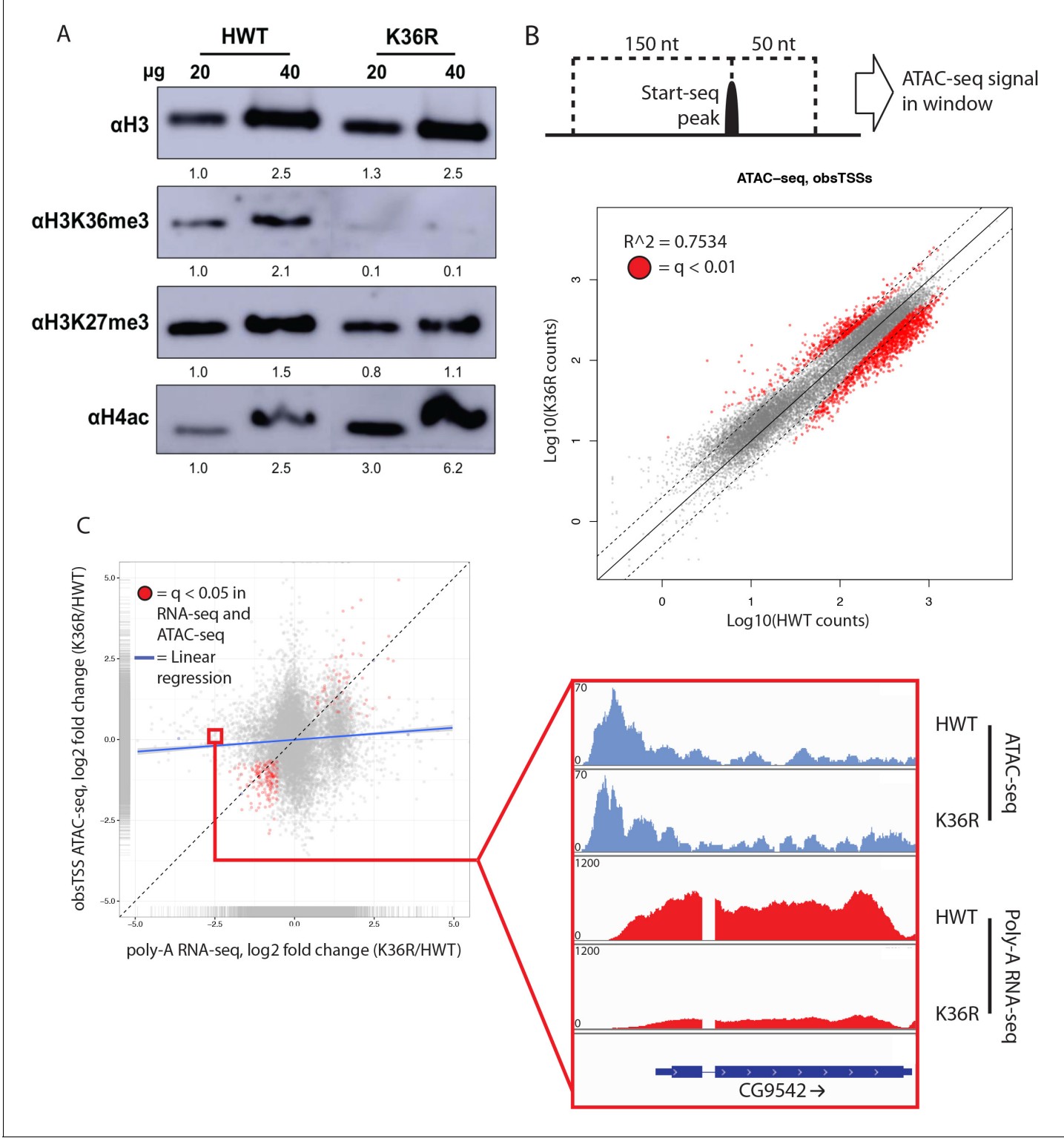

**Figure 3.** H4 acetylation enrichment in mutants does not result in open-chromatin-dependent changes in gene expression. (**A**) Western blots measuring enrichment of histone H3, H3K36me3, H3K27me3, and pan H4 acetylation (H4ac) in H3K36R mutants and HWT controls. Signal relative to first lane is denoted below each band. (**B**) Scatterplot of ATAC-seq signal mapping in a 200 nt window (as denoted at top) around obsTSSs, with $R^2$ value indicated. (**C**) Scatterplot of log2 fold change of poly-A RNA-seq (x-axis) vs. that of ATAC-seq (y-axis) signal in a window around the corresponding gene's transcription start site (as identified by start-seq). Genes with codirectional, statistically significant changes in both RNA-seq and

*Figure 3 continued on next page*

*Figure 3 continued*

ATAC-seq are indicated in red. Example browser shot of a gene differentially expressed in mutants in the absence of changes in chromatin accessibility at its start site is shown at right.

The following figure supplements are available for figure 3:

**Figure supplement 1.** Histone crosstalk and gene expression changes in H3K36R animals.

**Figure supplement 2.** Metagene analysis of Start-seq reads at previously annotated (observed) transcription start sites, obsTSSs.

We examined whether the position of a nuTSS relative to its closest annotated gene had any bearing upon changes in nuTSS usage in K36R mutants. Because exons are characterized by higher overall H3K36me3 signal than introns, they might be more sensitive to pervasive initiation. Furthermore, antisense initiation might also be more prevalent in the absence of H3K36me, as has been observed in budding yeast (*Carrozza et al., 2005*; *Keogh et al., 2005*). To test these ideas, we sorted nuTSSs by their position (exonic or intronic) and orientation (sense or antisense) relative to the resident gene. Analysis of modEncode ChIP-seq read density in 400 bp windows around each nuTSS confirmed that exonic nuTSSs are enriched for H3K36me3 relative to intronic ones (*Figure 4B*). Similarly, exonic nuTSSs are depleted of ATAC-seq open chromatin signal (*Figure 4—figure supplement 1A*).

Contrary to expectation, exonic and antisense nuTSS usage was not dramatically increased in K36R mutants (*Figure 4B*). Across all nuTSSs, we found that H3K36me3 density was anticorrelated with change in nuTSS 'usage,' that is, nuTSSs with lower signal in K36R than in HWT tended to have high H3K36me3 density, and vice-versa (*Figure 4—figure supplement 1B–E*). When we analyzed sense and antisense Start-seq reads mapping to annotated coding regions as a proxy for cryptic transcription in annotated genes, we found that antisense initiation did not globally accumulate in an H3K36me3-dependent manner (*Figure 4C*). These results show that modification of replication-dependent H3K36 is not required to suppress cryptic transcription in gene bodies. Instead, we found that pervasive initiation in gene bodies is widespread throughout the *Drosophila* genome, even in the presence of H3K36me.

We also studied the change in nuTSS usage relative to gene boundaries. When absolute change in Start-seq signal at each nuTSS is scaled to gene length, increased nuTSS usage occurs almost exclusively in intergenic regions (*Figure 4D*). Decreased usage is most prominent in the gene body, proximal to the 3' end (*Figure 4D*). Metagene analysis shows that these regions correlate with H3K36me3 ChIP-seq density (*Figure 2A*). Importantly, these findings do not support a role in *Drosophila* for H3K36me in suppressing cryptic antisense transcription, as described in yeast.

## H3K36 mutation does not affect alternative splicing

The H3K36me3 methyltransferase, SETD2, is reported to play a role in regulating alternative splice site choice (*Luco et al., 2010*; *Pradeepa et al., 2012*). To determine whether changes in pre-mRNA splicing contribute to gene expression differences between HWT and K36R, we used the MISO analysis package (*Katz et al., 2010*), which utilizes an annotated list of alternative splicing events, and quantitates changes between RNA-seq datasets. We found that very few annotated exon skipping events or retained intron events were significantly different between K36R and HWT, and there was no discernable bias toward inclusion or exclusion (*Figure 5A*). Additionally, the majority of high-confidence differential splicing events we detected were mild changes at best ($\Delta$PSI < 0.25), indicating that a lack of K36 modification had little effect on alternative splicing regulation in K36R mutants (*Figure 5B*).

Inappropriate intron retention is another class of splicing defect observed in SETD2 mutants (*Simon et al., 2014*). To examine intron retention events, we quantitated junction (j) and non-junction (n) reads mapping to every exon-exon boundary represented in our RNA-seq dataset. As shown in *Figure 5C*, we generated a retention ratio score (R) that measures the number of non-junction reads as a proportion of total reads (j+n). For junctions meeting statistical power requirements (>20 total reads), we observed no changes in the retention ratio, meaning that splice junction usage was

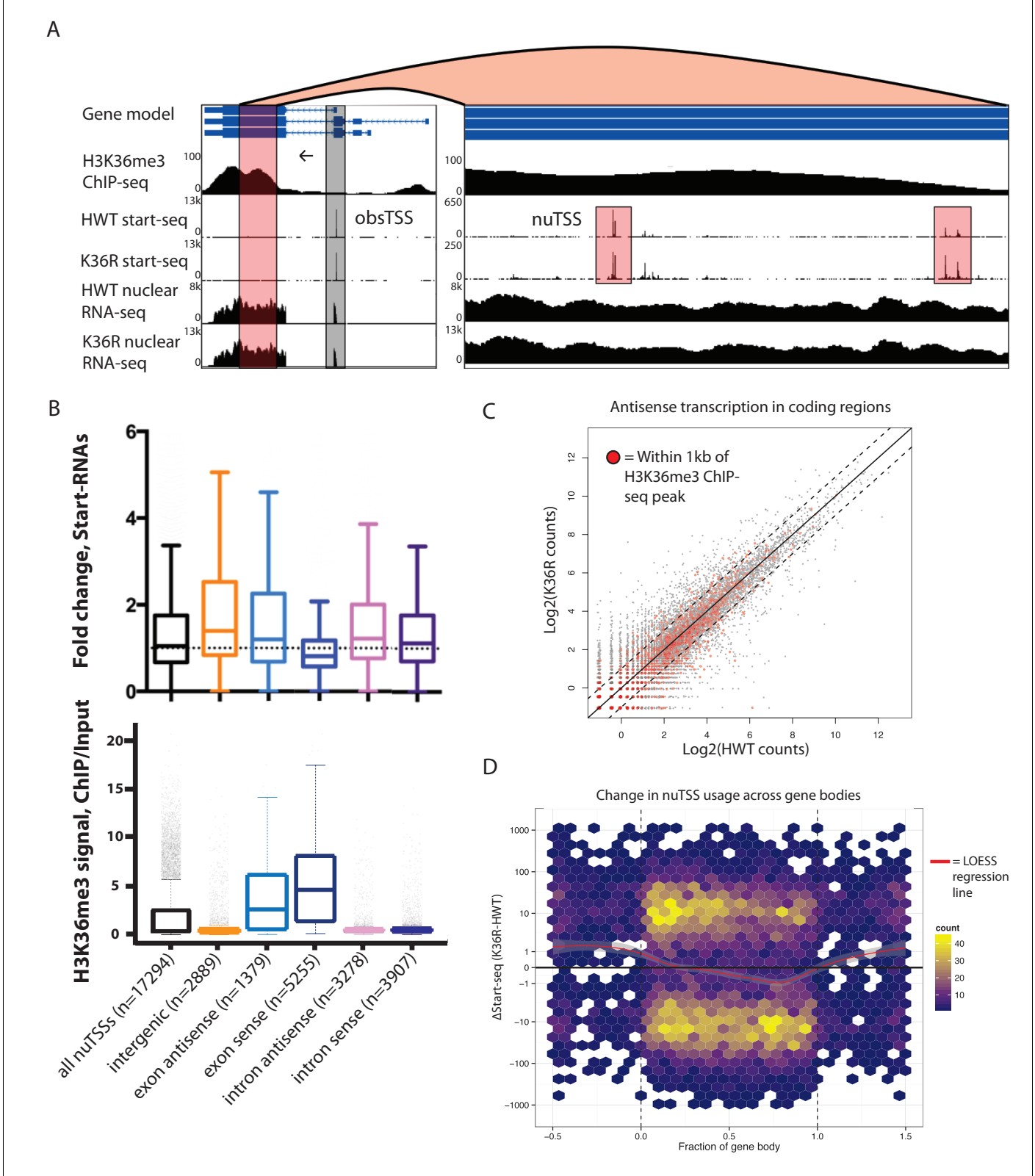

**Figure 4.** H3K36 modification does not suppress cryptic transcription initiation in coding regions. (**A**) Representative browser shot of gene containing novel unannotated transcription start sites (nuTSSs, highlighted in red). Direction of transcription denoted by arrow, and read counts denoted on Y-axis. (**B**) Boxplot describing the fold change in Start-seq signal for nuTSSs classified by their genomic localization and strand of origin relative to the resident gene if applicable. Lower boxplot describes H3K36me3 ChIP-seq signal (ChIP/input) for the same gene cohorts. (**C**) Scatterplot of normalized nuclear

*Figure 4 continued on next page*

*Figure 4 continued*

RNA-seq reads mapping antisense to genes in the dm3 reference gene model in HWT (x-axis) or K36R (y-axis). Genes containing or within 1 kb of a local H3K36me3-ChIP-seq peak are denoted by red dots. (D) Hex-plot heatmap plotting nuTSSs by their location relative to the gene boundaries of the nearest gene, and the absolute change in their Start-seq signal (K36R – HWT).

The following figure supplement is available for figure 4:

**Figure supplement 1.** Metagene analysis of Start-seq reads at novel, unannotated (nu)TSSs in comparison to open chromatin, nucleosome positioning and H3K36 trimethylation.

unchanged in K36R (*Figure 5C*). Taken together, these results support an H3K36me-dependent role for transcriptome regulation that is independent of alternative splicing.

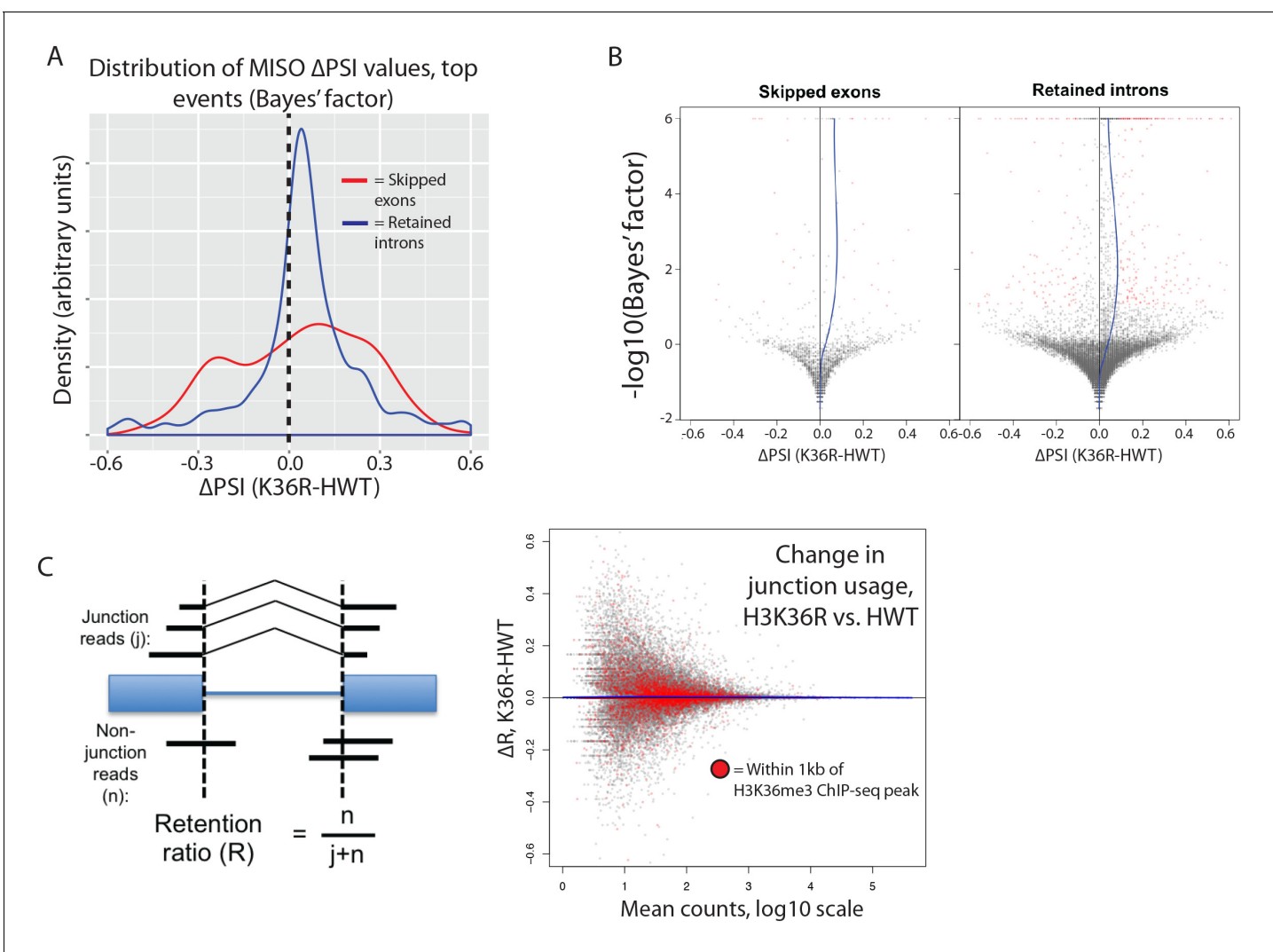

**Figure 5.** H3K36 modification does not regulate alternative splicing. (A) Density plots reflecting the distributions of change in percent spliced in (ΔPSI) values for skipped exon (red) or retained intron (blue) alternative splicing events manually classified as significant based on MISO parameters (see Materials and methods). (B) Volcano plots for skipped exon (left) and retained intron (right) events, with a local regression line (blue line) reflecting the skew in ΔPSI values (x-axis) based on Bayes factor (y-axis). (C) Global analysis of splice junction usage, where R denotes the 'retention ratio' in one condition, and ΔR denotes the difference in R between K36R and HWT.

## A class of highly expressed genes is under-represented in poly-A vs. nuclear RNA fractions due to sensitivity to exonuclease degradation

When comparing our poly-A and nuclear RNA-seq results, we identified a group of highly-expressed genes whose transcripts were reduced in the mutant poly-A RNA fraction but not in the corresponding nuclear RNA fraction (*Figure 6A*, see full RNA-seq results in *Supplementary file 1*). Transcripts identified in the nuclear RNA-seq data represent populations of newly transcribed as well as nuclear-retained RNAs, whereas poly-A selected RNA is thought to be comprised primarily of 'mature' mRNAs. We deduced that the observed differences between the two sequencing datasets could reflect a role for H3K36 in post-transcriptional, rather than co-transcriptional, mRNA maturation steps (e.g. nuclear RNA surveillance and export). Therefore, we selected a handful of mRNAs with large discrepancies between their nuclear and poly-A RNA-seq expression values (*Figure 6B*) for validation and testing by RT-PCR. Fractionation of nuclear and cytoplasmic RNA from HWT and K36R larvae prior to reverse transcription revealed no significant changes in subcellular localization of the targets (*Figure 6—figure supplement 1A*), suggesting that a global block to mRNA export due to H3K36R mutation is unlikely.

In the absence of a transport block, we surmised that mRNA surveillance and degradation pathways might contribute to the reduced transcript levels observed in the poly-A fraction. We therefore hypothesized that perturbation of RNA exonuclease activity might rescue target transcript levels by preserving immature mRNAs that would otherwise be degraded. We analyzed the effect on target mRNAs of depleting *Rrp6* and *Xrn1/pacman* (*pcm* in flies) by RNA interference (RNAi), utilizing the Gal4-UAS expression system (*Brand and Perrimon, 1993*). Flies sourced from the Transgenic RNAi Project (*Ni et al., 2011*) expressing short-hairpin (sh)RNA constructs and Gal4-drivers were crossed into the HWT and K36R genetic backgrounds. Unfortunately, RNAi for Rrp6 caused early larval lethality and animals of the appropriate genotype could not be obtained. However, we were able to introgress the Xrn1/pcm RNAi transgene into the HWT and K36R backgrounds and total RNA was prepared from whole third instar larvae. As shown in *Figure 6C*, the observed expression differences in poly-A RNA for a handful of highly expressed genes were restored to levels more similar to HWT in the K36R background by RNAi-mediated depletion of *pcm*. These results suggest that H3K36 contributes to post-transcriptional mRNA maturation in a manner that preserves target transcripts from exonuclease-mediated degradation.

## Defects in post-transcriptional processing contribute to gene expression changes in K36R mutants

mRNA degradation by *Xrn1/pcm* is preceded by two major surveillance steps: deadenylation by the CCR4-NOT complex, and decapping of the 7-methylguanosine (m7G) cap, primarily by the *Dcp2* decapping enzyme (*Sheth and Parker, 2003*). We therefore carried out RNAi against *CCR4/twin*, *CNOT7/Pop2*, and *Dcp2* in comparison with *Xrn1/pcm*. Across a panel of K36R downregulated genes, expression levels were rescued by depletion of *pcm*, *twin*, and *Pop2* (*Figure 6C*), but not by RNAi against *Dcp2* (*Figure 6—figure supplement 1B*). Given the known redundancies in decapping enzymes (e.g. see *Chang et al., 2012*), the negative results for the Dcp2 RNAi are inconclusive. Indeed, previous studies in S2 cells showed that depletion of Dcp2 alone is insufficient to effectively inhibit decapping (*Eulalio et al., 2007*). However, the positive results we obtained by depleting deadenylase factors led us to focus on polyadenylation.

Changes in 3' end formation and polyadenylation, which occur proximal to the H3K36me3-rich chromatin at the 3' ends of genes, might render mRNAs more sensitive to surveillance and degradation. To investigate this possibility, we analyzed poly-A tail length in the *CCR4/twin* RNAi background for a YFP reporter transgene using a modified LM-PAT assay (*Sallés et al., 1999*), as illustrated in *Figure 6—figure supplement 1C*. It is important to note that expression of the *UAS:YFP* transgene is directly tied to Gal4 expression and thus YFP is the only transcript that is guaranteed to be expressed in the same cells as the UAS:RNAi transgene. *UAS:YFP* is similarly sensitive to *pcm* and *twin* as our cohort of endogenous genes (*Figure 6—figure supplement 1D*), making it an ideal reporter. As shown in *Figure 6D*, we found that the YFP transcript displayed reduced poly-A tail length in K36R mutants, indicative of a role for H3K36 methylation over terminal exons in recruitment or functioning of the polyadenylation machinery. Importantly, the shorter poly-A tail in K36R mutants was independent of deadenylation activity (*Figure 6D*), demonstrating that the defect is in

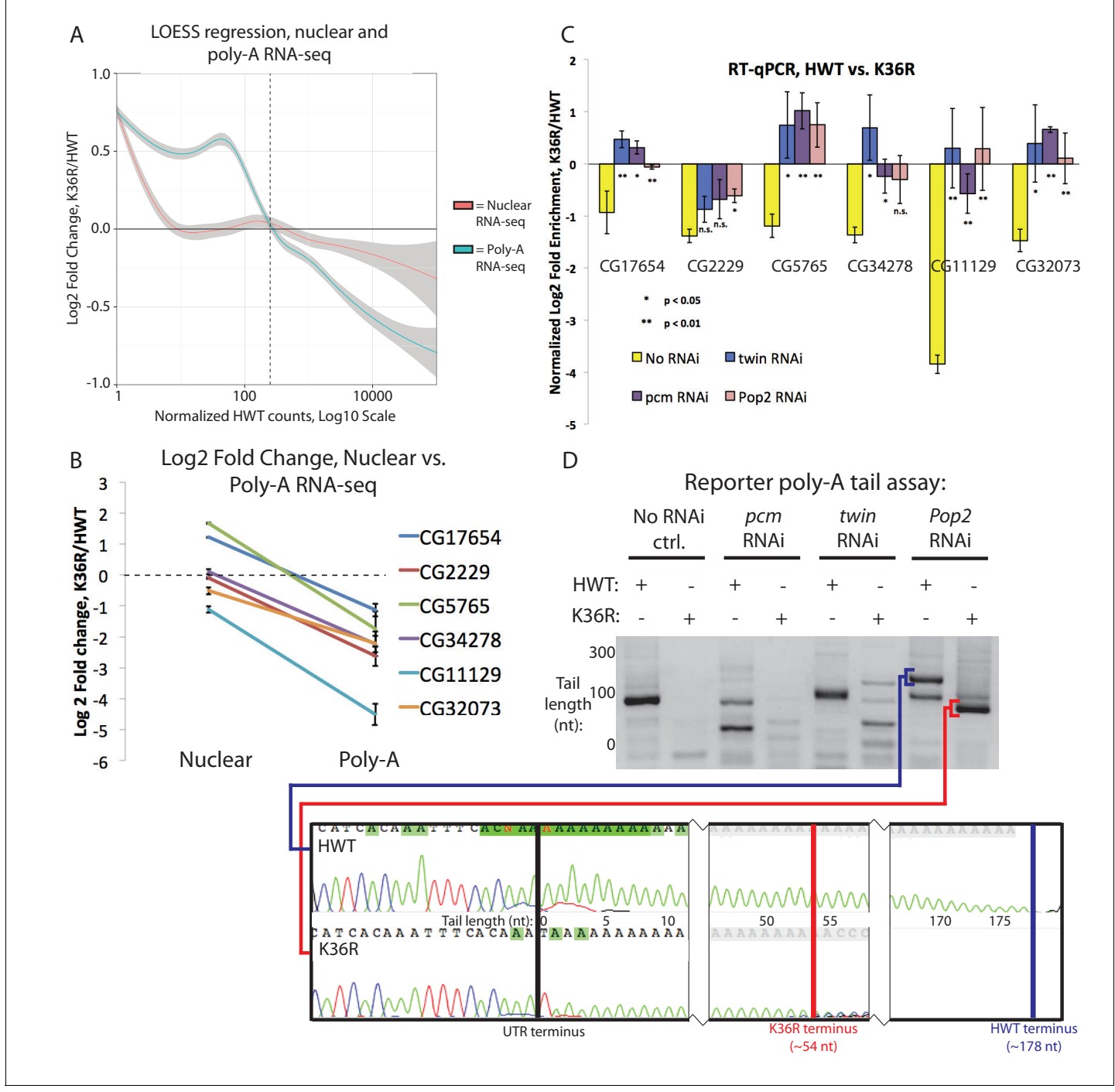

**Figure 6.** A class of highly expressed genes is subject to exonuclease degradation and inefficient post-transcriptional processing in H3K36R mutants. (A) LOESS regression lines generated from MA plots of either nuclear or poly-A RNA-seq, plotting gene log2 fold change (y-axis) vs. normalized read counts in HWT (x-axis). (B) Log2 fold change values between K36R and HWT in nuclear (left) and poly-A (right) RNA-seq, plotted for genes selected for further RT-PCR analysis. (C) RT-qPCR quantification of differential expression between HWT and K36R for select genes in a no RNAi, *pacman* RNAi, *twin* RNAi, or *Pop2* RNAi background, using the -ΔΔC$_t$ method. (D) LM-PAT assay results for the YFP transcript in HWT and K36R, in a no RNAi, *pcm* RNAi, *twin* RNAi, or *Pop2* RNAi background. Sanger sequencing trace confirming the poly-A site (leftmost panel) and differential tail lengths (right two panels) is shown below.

The following figure supplement is available for figure 6:

*Figure 6 continued on next page*

*Figure 6 continued*

**Figure supplement 1.** RT-PCR controls, alternative polyadenylation analysis and schematic of assay for gene-specific poly A tail length assay (LM-PAT) showing relative positions of primers.

polyadenylation, not in the subsequent CCR4/twin- or CNOT7/Pop2-dependent deadenylation. Additional experiments will be needed to determine the prevalence of poly-A tail length changes in the K36R mutants transcriptome wide. Computational analysis of differential poly-A site usage demonstrated no change (*Figure 6—figure supplement 1E*), indicating that poly-A site specification was largely unaffected by mutation of H3K36. In summary, these data uncover a post-transcriptional role for H3K36 in the regulation of metazoan gene expression.

## Discussion

In this study, we focus on the role of H3K36 in transcriptome fidelity, assayed at the levels of transcription initiation, elongation, pre-mRNA splicing and maturation. Crucially, most of the studies on the roles of H3K36me3 in animal cells deplete SETD2 or its orthologue, making it difficult to discern the specific role of the histone residue itself. Enzymes that catalyze histone PTMs often have numerous non-histone substrates or non-catalytic structural roles that can confound analysis (*Biggar and Li, 2015*; *Huang and Berger, 2008*; *Sims and Reinberg, 2008*; *Zhang et al., 2015*). Notably, alpha-tubulin was recently identified as a non-histone substrate of SETD2 (*Park et al., 2016*). Perhaps more importantly, SETD2 catalyzes trimethylation of lysine 36 in both the 'canonical' replication-dependent H3 and in the replication-independent histone variant, H3.3. H3.3 is thought to play a particularly important role in transcriptionally active regions where H3K36 methylation is enriched (*Ahmad and Henikoff, 2002*). Indeed, a protein with specific affinity for SETD2-catalyzed trimethylation of lysine 36 of the histone H3.3 variant was shown to serve as a regulator of RNA pol II elongation (*Wen et al., 2014*) and to associate with components of spliceosomal snRNPs to regulate co-transcriptional alternative mRNA splicing (*Guo et al., 2014*). Beyond its other substrates, SETD2's prominent association with the C-terminal domain of RNA pol II (*Kizer et al., 2005*) makes it likely that ablating this protein will result in transcriptional consequences that are unrelated to its catalytic activity. In view of these complications, the direct analysis of histone residue function enabled by our BAC-based gene replacement system is particularly well suited to the study of H3K36me in the context of transcriptional regulation.

In budding yeast, H3K36me2/3 has been shown to negatively regulate histone acetylation within actively transcribed genes, both by recruiting a repressive Rpd3S deacetylase complex (*Carrozza et al., 2005*; *Keogh et al., 2005*) and by suppressing incorporation of acetylated nucleosomes at sites of RNA polymerase II-initiated nucleosome displacement (*Venkatesh et al., 2012*). However, a similar role has not yet been elucidated for H3K36me in animals, and studies that have correlated cryptic transcription with H3K36 methylation in metazoan systems have done so only through perturbation of the SETD2 writer enzyme (*Carvalho et al., 2013*; *Xie et al., 2011*). Furthermore, studies have implicated H3K36me3 in alternative splicing in human cell culture (*Luco et al., 2010*; *Pradeepa et al., 2012*) and inefficient intron splicing in clear cell renal cell carcinomas (*Simon et al., 2014*), again via SETD2 mutation. In this study, we used histone replacement to define whether modification of canonical H3K36 is responsible for these functions.

We demonstrate that H3K36 is neither a significant contributor to the regulation of alternative splice site choice, nor the efficiency of canonical intron removal. We also present evidence that methylation of H3K36 does not suppress cryptic transcription in coding regions. Given the unprecedented depth of our Start-seq dataset (>200 M reads per genotype), even very rare events would have been detected. To the contrary, we found evidence for pervasive initiation (both sense and anti-sense) events that largely fail to appear in the steady-state RNA population under wild type conditions. Interestingly, we confirm that H4 acetylation is strongly suppressed by H3K36 modification, despite the fact that cryptic transcripts do not appear. This finding argues for an uncoupling of H4ac levels from cryptic initiation in coding regions in metazoans, and suggests that the suppression of cryptic transcription initiation in multicellular organisms may be more complex than previously appreciated.

One potential explanation for the discrepancy between our results and previous studies of SETD2 could be that modification of the aforementioned histone variant, H3.3, is the primary functional contributor to the cryptic initiation or splicing phenotypes. Elucidating the effects of H3.3K36 methylation is outside the scope of this work, and thus phenotypes that have been reported in the literature as being sensitive to H3K36 methylation might plausibly respond specifically to H3.3K36 methylation. In fact, this serves as a useful feature of histone replacement in this context, since a functional separation of H3 and H3.3 lysine 36 methylation cannot be otherwise achieved. However, this possibility should be tempered by the fact that we observed very low levels of H3K36me3 signal in both western blots from H3K36R mutant larvae (*Figure 3A*) and immunofluorescent staining of salivary gland polytene chromosomes (*McKay et al., 2015*). Thus H3.3 is, at best, a minor contributor to total H3K36me3. Future experiments testing the transcriptional consequences of direct mutation of H3.3K36, both on its own and in combination with mutation of replication-dependent H3K36, will better define their contributions.

Finally, we present evidence that H3K36 is required for proper mRNA maturation, providing a post-transcriptional benefit across a range of highly expressed genes. Additional studies will be required in order to elucidate a detailed molecular mechanism for this process. Our genetic suppression data suggest that this mRNA 'fitness' benefit is somehow linked to the efficiency of 3' end formation or polyadenylation (*Figure 6B–D*). Interestingly, H3K36me3 depletion in SETD2-mutant renal cell carcinoma has been correlated with defects in transcriptional termination and readthrough into neighboring genes (*Grosso et al., 2015*), suggesting that H3K36 methylation might influence termination and polyadenylation. Indeed, the enrichment of H3K36me3 at the 3' ends of genes makes it a likely candidate to interface with these activities. Another possibility is that H3K36 modification might recruit some type of RNA modifying enzyme. For example, Jaffrey and colleagues recently showed that dimethylation ($N^6$,2'-O-dimethyladenosine, or $m^6A_m$) of the nucleotide adjacent to the m7G cap enhances transcript stability (*Mauer et al., 2017*). Moreover, H3K36 might contribute to mRNA maturation across multiple processing steps, with the combined effect of protecting target mRNAs from surveillance and eventual degradation.

The prevailing model for histone PTM modulation of gene expression, reinforced by recent direct evidence (*Hilton et al., 2015*), suggests that it occurs directly proximal to the site of histone modification. However, the fact that genomic regions largely lacking H3K36me exhibit differential expression in K36R mutants argues against this idea. For that reason, a model for H3K36 control of gene expression should also consider indirect mechanisms. For example, it is possible that the rate of transcribing polymerase through nucleosomes that are modified at H3K36 might change, and therefore the capping, cleavage and polyadenylation machinery that associates with the C-terminal domain of RNA polymerase II (*Ho et al., 1998*; *McCracken et al., 1997*) might become improperly distributed in K36R mutants. Alternatively, SETD2 could have additional (unknown) substrates that function in these processes. Finally, H3K36me's previously reported role in three-dimensional genome organization (*Evans et al., 2016*; *Smith et al., 2013*; *Ulianov et al., 2016*) might extend to the concentration of factors related to mRNA maturation at sites of active transcription, which would be impaired upon H3K36 mutation. Future studies using alternative genetic approaches, including specific ablation of the catalytic activity of 'writers' to cross-reference our observations, should be instructive in this regard.

## Materials and methods

### RNA library preparation for sequencing

RNA-seq libraries were prepared using the Illumina TruSeq stranded library preparation kit from RNA prepared with TRIzol reagent (Thermo Fisher) from either whole third instar larvae (poly-A) or nuclei isolated from third instar larvae (nuclear) (adapted from [*Nechaev et al., 2010*]). Start-seq libraries were prepared as previously described (*Henriques et al., 2013*; *Nechaev et al., 2010*). Sequencing was carried out on a HiSeq2000 (ATAC-seq, poly-A and nuclear RNA-seq) or NextSeq500 (Start-seq) (Illumina). For all assays, at least three biological replicates were prepared (four in the case of Start-seq and nuclear RNA-seq).

## Start-seq

Total nuclear RNA from whole third instar larvae was used as input to each Start-seq library. For each RNA replicate used as input for a Start-seq library, 80 whole third instar larvae were collected. Five whole third instar larvae were selected for genomic DNA recovery via phenol chloroform extraction and ethanol precipitation in order to normalize Start-seq RNA spike-in controls to DNA content. The remaining (75) larvae were washed 3x with ice cold 1x ENIB buffer (15 mM Hepes pH7.6; 10 mM KCl; 3 mM CaCl$_2$; 2 mM MgCl$_2$; 0.1% Triton X-100; 1 mM DTT; 1 mM PMSF), and were then combined with 1 vol 0.3 M ENIB (1x ENIB +0.3 M Sucrose). Larvae were homogenized in a 1 mL dounce with 10 strokes with a type A pestle. Each replicate required douncing in three separate aliquots so as to avoid oversaturation of the dounce with larval cuticle, and homogenate was immediately transferred to ice once completed. Dounce was washed with 1 vol 0.3 M ENIB, combined with homogenate, and mixture was homogenized with 10 strokes with a type B pestle. Resulting homogenate was filtered through 40 µM Nitex mesh into a 50 mL conical tube on ice. For each 150 µL of filtered homogenate produced, a sucrose cushion was made by layering 400 µL 1.7 M ENIB followed by 400 µL 0.8 M ENIB in a 1.5 mL Eppendorf tube. 150 µL filtered homogenate was pipetted onto cushion, and spun at 20,000xg for 15 min at 4°C. After spinning, lipid residue was carefully removed from the walls of the tube with a micropippetor, and then the remainder of the supernatant was removed. The nuclear pellet was homogenized in 100 µL 0.3 M ENIB, and 10 µL was removed, stained with Trypan Blue, and observed under a microscope to confirm efficient nuclear isolation. Total RNA was extracted from the remaining homogenate with Trizol reagent using standard manufacturer's protocols. Start-seq libraries were prepared from nuclear RNA as previously described (*Henriques et al., 2013*; *Nechaev et al., 2010*). Libraries were sequenced on a NextSeq500 generating paired-end, 26 nt reads.

## Poly-A-selected RNA-seq

For each replicate, total RNA from 25 whole third instar larvae was isolated using Trizol reagent according to manufacturer's protocols. RNA-seq libraries were generated with the Tru-seq Stranded Poly-A RNA-seq library preparation kit (Illumina). Libraries were sequenced on a HiSeq2000 generating paired-end, 100 nt reads (Illumina).

## Nuclear RNA-seq

Nuclei from whole third instar larvae were isolated as described above for Start-seq, and RNA was extracted using Trizol reagent. Total nuclear RNA was used as input to Ribo-zero Stranded RNA-seq library preparation (Illumina). Libraries were sequenced on a HiSeq2000 generating paired-end, 50nt reads (Illumina).

## ATAC-seq library preparation

For each replicate, nuclei from 10 whole third instar larvae were isolated as per Start-seq and nuclear pellets were gently homogenized with wide-bore pipette tips in 50 ΔuL ATAC-seq lysis buffer (10 mM Tris·Cl, pH 7.4, 10 mM NaCl, 3 mM MgCl$^2$, 0.1% (v/v) Igepal CA-630), and homogenate was directly used as input to the Nextera DNA library preparation kit (NEB) for tagmentation of chromatinized DNA, as described in *Buenrostro et al. (2013)*. Libraries were sequenced on a HiSeq2000 generating single-end, 100 nt reads (Illumina).

## Bioinformatic analysis

Sequencing reads were mapped to the dm3 reference genome using Bowtie2 (*Langmead and Salzberg, 2012*) (ATAC-seq, Start-seq) or Tophat (*Trapnell et al., 2012*) (RNA-seq) default parameters. We used DESeq2 (*Love et al., 2014*) for differential expression analysis and Cufflinks (*Trapnell et al., 2012*) for novel transcript detection. We used the MISO package (*Katz et al., 2010*) to analyze annotated alternative splicing events, and custom scripts (*Source code 2*) to analyze global splice junction usage. Start-seq and ATAC-seq reads were mapped using Bowtie2 (*Langmead and Salzberg, 2012*), and Poly-A and nuclear RNA-seq reads were mapped using the Tophat gapped read aligner (*Trapnell et al., 2012*). Boxplots and Start-seq plots scaled to gene length were generated using ggplot2 in R (www.r-project.org).

For Start-seq, reads were quantified at base-pair resolution using a custom script (*Source code 1*), and nucleotide-specific raw read counts were normalized based on reads mapping to RNA spike-in controls. Exonic, intronic, and intergenic locations were determined using the dm3 gene model.

For Poly-A and nuclear RNA-seq: to analyze annotated alternative splicing, we used MISO (*Katz et al., 2010*), and considered splicing events with a) a Bayes score greater than 10 with all replicates combined, b) and consistent directionality of ΔPSI in each of the three individual replicates, as significant. To analyze global splice junction usage, we used a custom script (*Source code 2*) to quantify reads spanning the junction location that either map to it ('junction', i.e. containing an 'N' CIGAR designation that maps precisely to the junction in question) or through it ('non-junction'). To analyze differential expression, we used DESeq2 (*Love et al., 2014*) to quantify log2 fold change in normalized read counts between K36R and HWT. To analyze alternative polyadenylation, we used DaPars (*Xia et al., 2014*).

All ChIP-seq data were downloaded from modEncode (www.modencode.org). In all cases, data were derived from the third instar larval time point as determined by modEncode developmental staging procedures. For ChIP-seq and ATAC-seq, metagene plots were generated using the Deeptools package (*Ramírez et al., 2014*).

## Reverse transcription and PCR assays

RNA was isolated with TRIzol reagent as described above, and reverse transcription was performed using random hexamers and Superscript III (Invitrogen), according to the manufacturer's protocols. For semi-quantitative PCR analysis, products were run on a 2% agarose gel, and bands were quantified using ImageJ. For qPCR, Maxima SYBR Green/ROX qPCR Master Mix (Thermo Scientific) was used. All qPCR analyses are based on three biological replicates, plotted with standard error.

For semi-quantitative PCR, PCR reactions were prepared in biological triplicate using 2x Red Master Mix (Apex Bioscience), and targets were amplified for 35 cycles of PCR with a 95°C denaturation step, a 60°C annealing step, and a 72°C elongation step. Reactions were run on a 2% agarose gel with EtBr for 30 min at 90 V, and bands were imaged on a UV transilluminator (GE Healthcare) and quantified using ImageJ. For RT-qPCR, reactions were prepared in biological triplicate using Maxima SYBR Green/ROX qPCR Master Mix (Thermo Scientific), and fluorescence was monitored across 40 cycles in 96 well plate format.

For LM-PAT, 1 μg total RNA was incubated with 5 pmol preadenylated lmPAT anchor primer (ppApCAGCTGTAGCTATGCGCACCGAGTCAGATCAG) (adenylated using 5' DNA Adenylation Kit, NEB), and ligated with T4 RNA Ligase 2, truncated K227Q (NEB) using manufacturers protocol. Ligated RNA was reverse-transcribed with Superscript III (Life Technologies) using an lmPAT RT primer (GACTCGGTGCGCATAGCTACAGCTG). Resulting first-strand cDNA was PCR-amplified using gene-specific forward primers (see *Supplementary file 2*) paired with nested lmPAT RT primers that contain terminal thymidines (GTGCGCATAGCTACAGCTGTTTT). PCR conditions were as follows: a preliminary round consisted of 12 cycles in which the annealing step was decreased by one degree Celsius in each cycle from 71°C to 60°C (between 95°C and 72°C denaturation and elongation steps, respectively), followed by 18 additional cycles with an annealing temperature at 60°C. After completion of the first round, 2 μL PCR product was used as template for a second round of PCR amplification with 25 cycles and an annealing temperature at 60°C. For 'tail' measurement, template was amplified with a nested gene-specific forward primer and lmPAT nested RT reverse primer. For 'UTR' measurement, template was amplified with a nested gene-specific forward primer and a 'TVN' primer anchored at the 3' UTR terminus.

## Western blotting

For each replicate, nuclei from 30 whole third instar larvae were isolated as per Start-seq and homogenized in 50 μL Extraction Buffer (320 mM $(NH_4)_2SO_4$, 200 mM Tris HCl (pH 8.0), 20 mM EDTA, 10 mM EGTA, 5 mM $MgCl_2$, 1 mM DTT, 1x Protease Inhibitor Cocktail (Roche)). Mixture was spun at 15,000xg for 5 min at 4°C and supernatant was recovered and immediately used in polyacrylamide gel electrophoresis. Gel was transferred to PVDF membrane and incubated with rabbit anti-H3 (Abcam, ab1791), rabbit anti-H3K36me3 (Abcam, ab9050), mouse anti-H3K27me3 (Abcam, ab6002), or rabbit anti-H4ac (Active Motif, #39177) primary antibody overnight. We used ImageJ to quantify western blot band intensities, and calculated ratios of K36R/HWT intensity for each target

across two independent biological replicates. Student's t-test was used to obtain p-values for ratio comparisons between H3 and other targets.

## Immunofluorescence

Salivary gland polytene chromosome squashes were performed on third instar larvae as previously described (*McKay et al., 2015*), using rabbit anti-H4K12ac polyclonal primary antibody (Active Motif, #39165) overnight, followed by AlexaFluor 594 goat anti-rabbit secondary antibody (Thermo-Fisher) for two hours, then DAPI for 10 min.

## Acknowledgements

We thank the UNC High Throughput Sequencing Facility for library preparation and general expertise, the TRiP at Harvard Medical School (NIH/NIGMS R01-GM084947) for providing transgenic RNAi fly stocks used in this study, members of the UNC Histone Replacement Consortium for critical review of data and figures, and Kayla Peck for help with plotting. MPM was supported by an NIH predoctoral fellowship, F31-CA177088. This work was supported by the NIH Epigenomics Roadmap Project, R01-DA036897 (to AGM, BDS and RJD), and by the Intramural Research Program of the NIH (Z01-ES101987), National Institute of Environmental Health Sciences (to KA).

## Additional information

### Competing interests

KA: Reviewing editor, *eLife*. The other authors declare that no competing interests exist.

### Funding

| Funder | Grant reference number | Author |
| --- | --- | --- |
| National Institutes of Health | For use of Harvard TRiP lines, R01-GM084947 | Michael P Meers<br>A Gregory Matera |
| National Cancer Institute | Ruth L. Kirschstein Predoctoral Fellowship, F31-CA177088 | Michael P Meers |
| Office of the Director | Epigenomics Roadmap Project, R01-DA036897 | Brian D Strahl<br>Robert J Duronio<br>A Gregory Matera |
| National Institute of Environmental Health Sciences | Intramural Research Program, Z01-ES101987 | Karen Adelman |

The funders had no role in study design, data collection and interpretation, or the decision to submit the work for publication.

### Author contributions

MPM, Conceptualization, Data curation, Software, Formal analysis, Validation, Investigation, Methodology, Writing—original draft, Writing—review and editing; TH, Formal analysis, Validation, Investigation, Methodology; CAL, Software, Formal analysis, Validation; DJM, Conceptualization, Writing—review and editing; BDS, Conceptualization, Funding acquisition, Project administration, Writing—review and editing; RJD, Conceptualization, Resources, Funding acquisition, Project administration, Writing—review and editing; KA, Resources, Formal analysis, Supervision, Funding acquisition, Project administration, Writing—review and editing; AGM, Conceptualization, Resources, Formal analysis, Supervision, Funding acquisition, Investigation, Writing—original draft, Project administration, Writing—review and editing

### Author ORCIDs

Michael P Meers, http://orcid.org/0000-0003-3438-3938
Karen Adelman, http://orcid.org/0000-0001-5364-334X
A Gregory Matera, http://orcid.org/0000-0002-6406-0630

## Additional files

### Supplementary files

• Source code 1. Custom perl script used to extract base-specific TSS counts from a Start-seq read SAM file. Briefly, an input SAM file is parsed for strand orientation and mate pair status, with second mates thrown out, and the genomic coordinates of the first base in the first mate hashed with a running count of reads corresponding to that entry. An optional input removes TSSs whose accumulate counts fail to reach a user-defined threshold.

• Source code 2. Custom perl script used to detect splice junctions de novo from an input RNA-seq SAM file and quantify junction and non-junction read counts for each entry. Briefly, for each read spanning a splice junction (i.e. containing an 'N' SAM flag), intron coordinates are defined and hashed with a running count of reads corresponding to those coordinates, which are deposited in a BED file. Non-junction reads are subsequently determined by intersecting output BED file with the original SAM file using bedtools (*Quinlan and Hall, 2010*).

• Supplementary file 1. Excel file summarizing DESeq2 (*Love et al., 2014*) results from comparing K36R to HWT from total nuclear or poly-A RNA-seq. For each gene, and each experiment (nuclear and poly-A), there are listed values (from left to right) for mean counts, log2 fold change (K36R/HWT), log2 fold change standard error, test statistic, p-value, and adjusted p-value.

• Supplementary file 2. List of primers used for qPCR and LM-PAT assays (see methods).

### Major datasets

The following dataset was generated:

| Author(s) | Year | Dataset title | Dataset URL | Database, license, and accessibility information |
|---|---|---|---|---|
| Meers MP, Matera AG | 2017 | Histone gene replacement reveals a post-transcriptional role for H3K36 in maintaining metazoan transcriptome fidelity | https://www.ncbi.nlm.nih.gov/geo/query/acc.cgi?acc=GSE96922 | Publicly available at the NCBI Gene Expression Omnibus (accession no: GSE96922). |

The following previously published datasets were used:

| Author(s) | Year | Dataset title | Dataset URL | Database, license, and accessibility information |
|---|---|---|---|---|
| Elgin S | 2013 | H3K36me3 abcam L3 Nuc Input expt.2225 | https://www.ncbi.nlm.nih.gov/geo/query/acc.cgi?acc=GSM1147189 | Publicly available at the NCBI Gene Expression Omnibus (Accession no: GSM1147189) |
| Elgin S | 2013 | H3K36me3 abcam L3 Nuc Input expt.2226 | https://www.ncbi.nlm.nih.gov/geo/query/acc.cgi?acc=GSM1147190 | Publicly available at the NCBI Gene Expression Omnibus (Accession no: GSM1147190) |
| Elgin S | 2013 | H3K36me3 abcam L3 Nuc ChIP expt.2259 | https://www.ncbi.nlm.nih.gov/geo/query/acc.cgi?acc=GSM1147191 | Publicly available at the NCBI Gene Expression Omnibus (Accession no: GSM1147191) |
| Elgin S | 2013 | H3K36me3 abcam L3 Nuc ChIP expt.2260 | https://www.ncbi.nlm.nih.gov/geo/query/acc.cgi?acc=GSM1147192 | Publicly available at the NCBI Gene Expression Omnibus (Accession no: GSM1147192) |

| | | | | |
|---|---|---|---|---|
| Elgin S | 2013 | H3K36me1 L3 Nuc Input expt.2402 | https://www.ncbi.nlm.nih.gov/geo/query/acc.cgi?acc=GSM1147193 | Publicly available at the NCBI Gene Expression Omnibus (Accession no: GSM1147193) |
| Elgin S | 2013 | H3K36me1 L3 Nuc Input expt.2404 | https://www.ncbi.nlm.nih.gov/geo/query/acc.cgi?acc=GSM1147194 | Publicly available at the NCBI Gene Expression Omnibus (Accession no: GSM1147194) |
| Elgin S | 2013 | H3K36me1 L3 Nuc ChIP expt.2400 | https://www.ncbi.nlm.nih.gov/geo/query/acc.cgi?acc=GSM1147195 | Publicly available at the NCBI Gene Expression Omnibus (Accession no: GSM1147195) |
| Elgin S | 2013 | H3K36me1 L3 Nuc ChIP expt.2401 | https://www.ncbi.nlm.nih.gov/geo/query/acc.cgi?acc=GSM1147196 | Publicly available at the NCBI Gene Expression Omnibus (Accession no: GSM1147196) |
| Elgin S | 2013 | H3 antibody3 L3 Nuc Input expt.2222 | https://www.ncbi.nlm.nih.gov/geo/query/acc.cgi?acc=GSM1147289 | Publicly available at the NCBI Gene Expression Omnibus (Accession no: GSM1147289) |
| Elgin S | 2013 | H3 antibody3 L3 Nuc Input expt.2224 | https://www.ncbi.nlm.nih.gov/geo/query/acc.cgi?acc=GSM1147290 | Publicly available at the NCBI Gene Expression Omnibus (Accession no: GSM1147290) |
| Elgin S | 2013 | H3 antibody3 L3 Nuc ChIP expt.2241 | https://www.ncbi.nlm.nih.gov/geo/query/acc.cgi?acc=GSM1147291 | Publicly available at the NCBI Gene Expression Omnibus (Accession no: GSM1147291) |
| Elgin S | 2013 | H3 antibody3 L3 Nuc ChIP expt.2242 | https://www.ncbi.nlm.nih.gov/geo/query/acc.cgi?acc=GSM1147292 | Publicly available at the NCBI Gene Expression Omnibus (Accession no: GSM1147292) |
| Karpen G | 2013 | H3K36me2 W 14-16 hr OR Emb Input expt.2307 | https://www.ncbi.nlm.nih.gov/geo/query/acc.cgi?acc=GSM1147547 | Publicly available at the NCBI Gene Expression Omnibus (Accession no: GSM1147547) |
| Karpen G | 2013 | H3K36me2 W 14-16 hr OR Emb Input expt.2308 | https://www.ncbi.nlm.nih.gov/geo/query/acc.cgi?acc=GSM1147548 | Publicly available at the NCBI Gene Expression Omnibus (Accession no: GSM1147548) |
| Karpen G | 2013 | H3K36me2 W 14-16 hr OR Emb ChIP expt.2396 | https://www.ncbi.nlm.nih.gov/geo/query/acc.cgi?acc=GSM1147549 | Publicly available at the NCBI Gene Expression Omnibus (Accession no: GSM1147549) |
| Karpen G | 2013 | H3K36me2 W 14-16 hr OR Emb ChIP expt.2397 | https://www.ncbi.nlm.nih.gov/geo/query/acc.cgi?acc=GSM1147550 | Publicly available at the NCBI Gene Expression Omnibus (Accession no: GSM1147550) |
| Elgin S | 2013 | H4K16ac(M).L3 Input expt.2402 | https://www.ncbi.nlm.nih.gov/geo/query/acc.cgi?acc=GSM1200107 | Publicly available at the NCBI Gene Expression Omnibus (Accession no: GSM1200107) |
| Elgin S | 2013 | H4K16ac(M).L3 Input expt.2404 | https://www.ncbi.nlm. | Publicly available at |

| | | | nih.gov/geo/query/acc. cgi?acc=GSM1200108 | the NCBI Gene Expression Omnibus (Accession no: GSM1200108) |
|---|---|---|---|---|
| Elgin S | 2013 | H4K16ac(M).L3 ChIP expt.2514 | https://www.ncbi.nlm. nih.gov/geo/query/acc. cgi?acc=GSM1200109 | Publicly available at the NCBI Gene Expression Omnibus (Accession no: GSM1200109) |
| Elgin S | 2013 | H4K16ac(M).L3 ChIP expt.2515 | https://www.ncbi.nlm. nih.gov/geo/query/acc. cgi?acc=GSM1200110 | Publicly available at the NCBI Gene Expression Omnibus (Accession no: GSM1200110) |

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
