## [Decision Letter]

Thank you for submitting your article "Histone gene replacement reveals a post-transcriptional role for H3K36 in maintaining metazoan transcriptome fidelity" for consideration by *eLife*. Your article has been favorably evaluated by Jessica Tyler (Senior Editor) and three reviewers, one of whom is a member of our Board of Reviewing Editors. The reviewers have opted to remain anonymous.

The results presented in this manuscript are novel and of great interest to the large field of gene expression regulation. Furthermore, the manuscript reveals that the phenotypes observed when the "writer" enzymes are depleted differ from the phenotypes observed when histone variants that cannot be modified are expressed.

The reviewers have discussed the reviews with one another and the Reviewing Editor has drafted this decision to help you prepare a revised submission.

Summary:

This manuscript investigates the apparently direct effects on transcription and RNA levels in a *Drosophila* mutant strain lacking H3K36 methylation. Contrary to other studies done in multicellular eukaryotes, the present study uses a K36R mutant where modification at this position is not possible. In that way, a direct function of the H3K36 PTM may be analyzed, whereas in strains lacking the writer enzyme, observed effects can derive from an additional function of SETD2, which is not its catalytic function, or from modification of other substrates. The authors use genome-wide analysis to show a global change in expression. Expression changes anti-correlate with levels of H3K36 methylation. The authors confirm the previously reported link between H3K36me and H4ac. They use ATAC-seq to investigate whether increased H4ac in K36 mutants led to higher chromatin accessibility and they did not find this. Instead they found – using Start-seq -novel unannotated TSSs (here called nuTSSs) in both HWT and K36 mutants. However, these nuTSSs were not restricted to genes containing K36me, suggesting that cryptic transcription occurs independently of H3K36me. Thus, this study does not support a role for H3K36me in suppressing cryptic transcription in *Drosophila*, as described in yeast. Another not supported function for H3K36me in this study, which was described before, is its role in alternative splicing. Differential splicing events, intron retention and exon exclusion events were hardly observed between the HWT and the K36R mutant.

Finally, the authors propose a role for H3K36me in post-transcriptional, mRNA maturation steps based on Poly-A-seq analysis showing downregulation of some genes in K36R mutants compared to HWT. These RNA levels were restored to more similar levels as in the HWT after RNAi of the exonuclease Xrn1. RNA levels were also restored when the deadenylation enzyme twin was depleted. In summary, this important study of H3K36 modification shows differences between yeast and multicellular eukaryotes and on the other hand addresses previous implicated roles for H3K36 methylation. The author propose a model that H3K36me maintains transcriptome fidelity, although it remains unclear how this is achieved.

Essential revisions:

1) Figure legends are lacking for the main figures and must be included.

2) It is unknown whether arginine really behaves like lysine on H3 in position 36. The R guanidinium head group enables different hydrogen bonding stoichiometries than the lysine head group. This must be mentioned in the beginning of the paper and properly discussed. I understand it is probably the best one can do, yet one cannot exclude secondary effects – although the primary effect of this mutation is very likely due to an inability to neutralize the positive charge of the side chain by methylation.

3) Figure 3: one can see that the 4 bands in the western blots for both H3K36me and H4ac come from different pictures. Please provide the original pictures. Additionally, H4ac ChIP-seq data in the K36R mutant would strengthen the results.

4) LM-PAT experiment: what are the two bands when using HWT and Tail primers? If deadenylation worked you shouldn't get any product. Or if the deadenylation was not completed, you should see lower bands for the mutant.

5) Please show more examples in which the pA site changes in K36K mutants or change the conclusions to reflect the fact that only one instance was observed.

6). The authors could comment on how H3K36me3 could protect mRNAs from degradation and what is the link between this phenotype and the short poly(A) tails.

7) Additionally, it is very unclear how the mRNAs are degraded. XRN1 cannot degrade capped mRNAs. On the other hand, the authors show that DCP2-depletion did not prevent degradation. Based on this observation they suggest that the mRNAs may be uncapped, but they were not able to show that this indeed the case. The explanation is probably much simpler: previous studies have shown that DCP2 depletion is not sufficient to prevent mRNA degradation in *Drosophila* cells and that at least two decapping factors should be codepleted to effectively block decapping. In other words, the author cannot rule out that the residual amounts of DCP2 in the RNAi knockdown are sufficient to sustain decapping; this possibility should be mentioned and discussed. In particular, because the efficacy of the DCP2 depletion has not been tested by western blot. Essentially, it is very likely that the mRNAs are degraded by deadenylation-dependent decapping (i.e., the 5'-to-3' decay pathway).

---

## [Author Response]

Essential revisions:

1) Figure legends are lacking for the main figures and must be included.

In the first submission, our figures and legends were included but they were in separate sections. Presumably to facilitate reading of manuscripts online, the *eLife* house style is that we should interleave the legends together with the figures. In the revision, we now include the legends with their corresponding figures.

2) It is unknown whether arginine really behaves like lysine on H3 in position 36. The R guanidinium head group enables different hydrogen bonding stoichiometries than the lysine head group. This must be mentioned in the beginning of the paper and properly discussed. I understand it is probably the best one can do, yet one cannot exclude secondary effects – although the primary effect of this mutation is very likely due to an inability to neutralize the positive charge of the side chain by methylation.

In both the Introduction and the Discussion sections, we revised the text to illustrate the potential complexities inherent in interpreting phenotypes derived from a K to R mutation, including potential neomorphic effects. We note, however, that methylation of lysine does not neutralize its positive charge, although bulky methyl groups certainly have the potential to occlude effector proteins from being able to form strong ionic bonds with it. Arginine maintains this positive charge but is chemically distinct, as mentioned above. In any event, this point is well taken and we are happy to spend a few more words in the Introduction to help readers interpret the findings.

3) Figure 3: one can see that the 4 bands in the western blots for both H3K36me and H4ac come from different pictures. Please provide the original pictures. Additionally, H4ac ChIP-seq data in the K36R mutant would strengthen the results.

Figure 7 shows the original images of the westerns shown in Figure 3. Equal amounts of each sample were divided onto four gels, and probed separately. We have found that stripping and re-probing these blots is not a good option. For the revision, we ran another set of four blots and quantified the combined results in Figure 3—figure supplement 1.

Author response image 1.**DOI:**
http://dx.doi.org/10.7554/eLife.23249.018

As for the addition of H4ac ChIP-seq data from the mutants, we agree that such an experiment might strengthen our conclusions insofar as it would enable us to identify the specific genomic regions where H4ac is increased in K36R animals, and subsequently correlate those data with changes in open chromatin and transcription initiation. We feel that such a detailed analysis is beyond the scope of the current manuscript (and beyond our current resources as well). However, in the revision we included new immunofluorescence data from salivary gland polytene squashes from K36R larvae to help address this question (see new Figure 3—figure supplement 1). In both HWT and K36R animals, H4K12ac localization is anticorrelated with the DAPI bright regions, which are thought to correspond to more transcriptionally silent chromatin. Therefore, the overall hyperacetylation in K36R mutants occurs in more actively transcribed regions. These findings are consistent with those of previous studies showing that H4ac accumulates in coding regions upon H3K36me3 depletion (Keogh et al., 2005; Carrozza et al., 2005).

4) LM-PAT experiment: what are the two bands when using HWT and Tail primers? If deadenylation worked you shouldn't get any product. Or if the deadenylation was not completed, you should see lower bands for the mutant.

We presume this is in reference to Figure 6 (HWT, tail primers, No RNAi control, see lane 5 in Figure 8). The lower band in the No RNAi control lane is inconsistent, but could represent two distinct populations of mRNA, one that has a longer polyA tail and one that is very short. The mRNA corresponding to the upper band would represent the translationally active population. Presumably, such a population does not appear in the (K36R, tail primers, No RNAi control) because most of these transcripts have already been targeted for degradation by *Xrn1/pacman*.

In the revision, we performed the LM-PAT assay again, this time adding data for Pop2 and *pcm* in addition to twin. To save space, we show only PCR products from the tail primers in the new Figure 6, although we used the data from the UTR primers to set the zero tail length. Similar to the previous results we obtained for the twin RNAi experiment (not shown in the current version of the manuscript but see sequence traces in Figure 8), we sequenced the PCR products from the *Pop2* RNAi lane and found that the HWT tails are again demonstrably longer than those for K36R. We therefore conclude that the observed differences in poly-A tail length between the mutant and wildtype reporter mRNAs is independent of deadenylation activity.

Author response image 2.**DOI:**
http://dx.doi.org/10.7554/eLife.23249.019

5) Please show more examples in which the pA site changes in K36K mutants or change the conclusions to reflect the fact that only one instance was observed.

To be clear, the analysis of pA site usage is presented in Figure 6—figure supplement 1. In the main Figure 6, we analyzed pA tail length. Our focus on the *UAS:YFP* reporter stems from the fact that this is the only transcript where we can be sure that the Gal4 driven RNAi construct is expressed in the exactly the same cells where the YFP reporter is expressed. To bolster our conclusions regarding the presumptive deadenylation by CCR4/twin, we carried out RNAi against another component of the CCR4-Not complex, *CNOT7/Pop2*. The new qRT-PCR results are shown in Figure 6. See also the response to point #7 below where we added a whole new set of RNAi experiments and tested the expression of endogenous genes.

The LM-PAT assay involves numerous cycles of (nested) PCR, and is thus somewhat finicky. We were unable to reliably PCR the pA tails of the other transcripts that we tested. We therefore amended the text to limit the conclusions as suggested.

6) The authors could comment on how H3K36me3 could protect mRNAs from degradation and what is the link between this phenotype and the short poly(A) tails.

We expanded the Discussion to outline a model for how K36me3 could promote overall transcript fitness by recruiting stabilization factors. Such factors could include the polyadenylation machinery as well as RNA methyltransferases. For example, one of the most prevalent modified bases is found at the 5′ end of an mRNA, on the first encoded nucleotide adjacent to the 7-methylguanosine cap. New work from the Jaffrey lab has shown that this modification, *N*^6^,2′-*O*-dimethyladenosine (m^6^Am), is reversible and influences mRNA stability (Mauer et al., 2017; doi:10.1038/nature21022).

7) Additionally, it is very unclear how the mRNAs are degraded. XRN1 cannot degrade capped mRNAs. On the other hand, the authors show that DCP2-depletion did not prevent degradation. Based on this observation they suggest that the mRNAs may be uncapped, but they were not able to show that this indeed the case. The explanation is probably much simpler: previous studies have shown that DCP2 depletion is not sufficient to prevent mRNA degradation in Drosophila cells and that at least two decapping factors should be codepleted to effectively block decapping. In other words, the author cannot rule out that the residual amounts of DCP2 in the RNAi knockdown are sufficient to sustain decapping; this possibility should be mentioned and discussed. In particular, because the efficacy of the DCP2 depletion has not been tested by western blot. Essentially, it is very likely that the mRNAs are degraded by deadenylation-dependent decapping (i.e., the 5'-to-3' decay pathway).

We agree with the referees that the whole decapping issue is inconclusive and that the mRNAs are almost certainly degraded using the normal 5’-to-3’ pathway. As pointed out in the critique, our RNAi-mediated knockdown of Dcp2 may not be that efficient in vivo. Or it could be that residual amounts of the enzyme are sufficient for activity. And, as also pointed out by the referees, there are known genetic redundancies (including Dcp2 paralogs) that could complicate the analysis.

Decapped mRNAs are very low abundance and short lived, so it is hard to quantify differences in levels of uncapped messages in two different samples by ligation/PCR-based methods. Because negative results are often not meaningful, we decided to remove the *Dcp2* RNAi analysis from the main figures and focus instead on shoring up our findings with the deadenylase complex. In the revision, we added a new set of experiments to Figure 6 wherein we analyze the effects of depleting a second member of the CCR4-Not complex, Pop2. We also re-did all of the qRT-PCR experiments (targeting *Pop2/CNOT7, twin/CCR4* and *pacman/XRN1*) by crossing the HWT and K36R transgenic lines onto a different Gal4 driver line (*armadillo*:Gal4), whose expression pattern is more ubiquitous than that of *twist*:Gal4. The results are very consistent.